# A living vector field reveals constraints on galactose network induction in yeast

Sarah R Stockwell & Scott A Rifkin*

## Abstract

When a cell encounters a new environment, its transcriptional response can be constrained by its history. For example, yeast cells in galactose induce GAL genes with a speed and unanimity that depends on previous nutrient conditions. Cellular memory of long-term glucose exposure delays GAL induction and makes it highly variable with in a cell population, while other nutrient histories lead to rapid, uniform responses. To investigate how cell-level gene expression dynamics produce population-level phenotypes, we built living vector fields from thousands of single-cell time courses of the proteins Gal3p and Gal1p as cells switched to galactose from various nutrient histories. We show that, after sustained glucose exposure, the lack of these GAL transducers leads to induction delays that are long but also variable; that cellular resources constrain induction; and that bimodally distributed expression levels arise from lineage selection—a subpopulation of cells induces more quickly and outcompetes the rest. Our results illuminate cellular memory in this important model system and illustrate how resources and randomness interact to shape the response of a population to a new environment.

**Keywords** dynamics; galactose; gene expression; single-cell; yeast

**Subject Categories** Quantitative Biology & Dynamical Systems; Transcription

**Mol Syst Biol. (2017) 13: 908**

## Introduction

Budding yeast cells (*Saccharomyces cerevisiae*) can metabolize galactose by inducing a network of regulatory and metabolic genes, collectively known as the GAL genes (Fig EV1). When activated by galactose that has been imported into the cell, Gal3p blocks the repressor Gal80p from inhibiting the action of the transcription factor Gal4p. Gal4p, in turn, promotes the transcription of the GAL genes, including the regulatory genes *GAL3, GAL1,* and *GAL80,* the membrane-bound galactose importer gene *GAL2,* and the enzymes *GAL1, GAL7,* and *GAL10* (Bram *et al*, 1986; Giniger & Ptashne, 1988; Lohr *et al*, 1995; Bryant & Ptashne, 2003; Abramczyk *et al*, 2012). The paralogs *GAL1* and *GAL3* transduce the galactose signal,

acting as positive regulators of the system (Abramczyk *et al*, 2012), and the network's interlocking positive and negative regulatory feedback loops control induction in the presence of galactose (Lohr *et al*, 1995; Acar *et al*, 2005; Ramsey *et al*, 2006). Abundant glucose represses GAL network activation (Nehlin *et al*, 1991; Johnston & Carlson, 1992; Bryant *et al*, 2008).

The GAL network has been an important model system for metabolism, gene regulation, and now quantitative biology for most of a century, and the behavior of this network in various carbon sources at steady state is well understood (Lohr *et al*, 1995; Braun & Brenner, 2004). However, induction time courses have revealed that the transient induction dynamics of the GAL network depend on cellular memory of previous nutrient environments (Stockwell *et al*, 2015). Cells previously grown in non-inducing/non-repressing media like raffinose or glycerol induce quickly and fairly uniformly (Johnston *et al*, 1994; Lohr *et al*, 1995) (Fig EV2). The same is true for reinducing cultures: cells that have undergone prior galactose induction followed by short-term, 12-h, glucose repression (Zacharioudakis *et al*, 2007) before being switched back to pure galactose (Figs 1, 2, and EV2; Movie EV1). By contrast, cell populations that have experienced long-term glucose repression (LTGR) induce the GAL genes after a long lag, producing a transiently bimodal distribution that, in population-level experiments, gradually resolves into an entirely induced population over the course of many hours (Fig EV2) (Biggar & Crabtree, 2001; Zacharioudakis *et al*, 2007). The transducer/enzyme Gal1p is required for the reinduction phenotype (Zacharioudakis *et al*, 2007), but the mechanisms and population biology behind the other memory phenotypes—particularly the transient bimodality after LTGR—are unknown, in part because research on GAL memory has largely been based on population-level measurements or snapshots of a population at a few times. As we show, such population-level measurements conflate the effects of growth and induction and mask the potential for competition between-cell lineages to reshape the composition of the cell population (Nevozhay *et al*, 2012).

## Results

The key to GAL network memory phenotypes may lie in the levels of the transducers Gal1p and Gal3p that cells possess when they

---

Section of Ecology, Behavior, and Evolution, Division of Biological Sciences, University of California, San Diego, La Jolla, CA, USA
*Corresponding author. Tel: +1 858 822 5748; E-mail: sarifkin@ucsd.edu

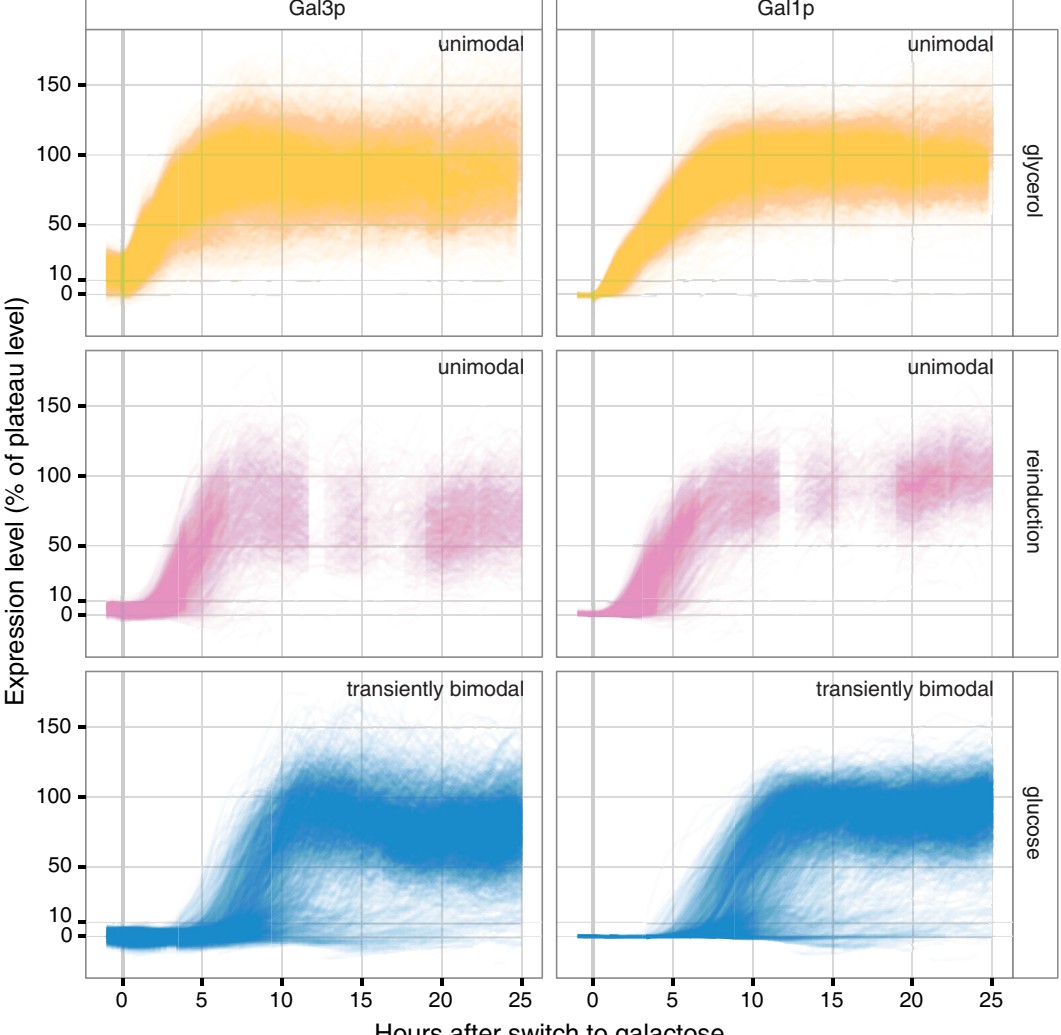

**Figure 1.  Cells induce quickly and uniformly after glycerol (yellow) and reinduction (pink), but variably after a long lag after long-term glucose repression (blue) producing a transiently bimodal population distribution.**

Single-cell time courses for each of the three history conditions for Gal3p-yECitrine and Gal1p-yECerulean. Expression levels have been normalized to an estimated 100% plateau level (see Materials and Methods). Each thin line corresponds to a single cell tracked over time, and each plot depicts trajectories from thousands of cells. Data points are spaced every 20 min, and photobleaching is minimal (Appendix Fig S4). Absolute Gal1p-yECerulean fluorescence levels are approximately 10× higher than Gal3p-yECitrine, reflecting the massive induction of Gal1p. Fluorophore maturation time is expected to be on the order of 30 min. The y-axis represents percent of plateau-level gene expression (see Materials and Methods). 0% represents the median of the control cell fluorescence levels for a given frame (i.e. no fluorescent proteins).

encounter galactose: cells accumulate different amounts of Gal1p and Gal3p in different nutrient environments (Appendix Table S1). Cells that already have a small amount of transducer when transferred to galactose media induce the network quickly and uniformly, while cells that lack either transducer exhibit a long lag and a bimodal induction pattern. These observations suggest that initial concentrations of Gal1p and Gal3p may explain much of the transient population level. The GAL regulatory feedback loops include other important components (Fig EV1), such as the galactose permease Gal2p which dramatically increases galactose uptake above the low levels provided by a constitutive facilitated diffusion process (Ramos *et al*, 1989) and Gal80p which acts as a negative regulator. However, Gal3p has long been recognized as critical to induction speed (*gal3* mutants take days to induce the network

instead of hours) (Winge & Roberts, 1948; Spiegelman *et al*, 1950; Douglas & Pelroy, 1963; Bhat & Hopper, 1991), and Gal1p is responsible for fast, unimodal induction upon reinduction (Zacharioudakis *et al*, 2007). Thus, the positive feedback loop between Gal1p and Gal3p (Fig EV1) may well govern the tempo and mode of GAL induction. In this scenario, a cell beginning with low concentrations of Gal1p/Gal3p would increase those proteins slowly, because neither transducer is abundant enough to ramp up GAL induction quickly. A cell with a moderate concentration of either protein will increase GAL expression quickly, because the positive feedback enables either transducer to accelerate the induction process for both. Finally, a cell approaching the equilibrium point where protein synthesis balances decay will begin to plateau and change protein concentrations slowly.

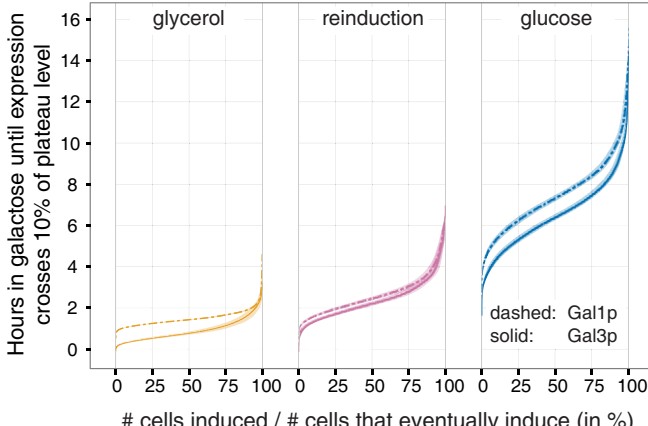

**Figure 2. Empirical cumulative distributions for each history condition for Gal3p and Gal1p.**
Only cells that started with expression levels below 10% and induced to at least 75% of the plateau level are included (in the glucose condition in particular, most cells fail to induce). Shaded areas denote 95% confidence intervals from bootstrapping.

To visualize the transient dynamics after the different media histories, we represent them as flows across a vector field on the state space of the two transducers, Gal1p and Gal3p (Figs 3 and EV3). A vector field is a standard tool for analyzing dynamical systems. It presents an efficient geometric summary of the overall behavior of a system in terms of key variables. Here, we translate this tool into a biological reality, generating *living* vector fields to summarize our measurements of thousands of individual cells tracked over time (Fig 1) and giving us a comprehensive view of their induction dynamics. In these vector fields, each vector illustrates how Gal1p and Gal3p concentrations change over a given time interval. The root of a vector represents the protein concentrations at some time, the direction points toward the concentrations at the next time point, and the length is proportional to its speed. By measuring the concentration of Gal3p and Gal1p in a single cell, we can place it at a particular point in the Gal3p/Gal1p state space (Appendix Fig S1). Because its movement through this state space depends solely on its current location and the direction of the vector field at that location, over time the protein concentrations in the cell will follow a trajectory described by the arrows starting from that point until they reach their steady-state levels where the arrow lengths shrink to zero (Appendix Fig S2). In the absence of noise or other causal variables, any cell proceeding from that same protein concentration would trace the same path. In this paper, we measured Gal3p and Gal1p levels by fusing 2x-yECitrine to Gal3p and yECerulean to Gal1p (Appendix Fig S3). We used a microfluidic device (Ferry *et al*, 2011) to measure GAL network induction at 20-min intervals (Appendix Fig S4) as we switched cells with fluorescently labeled Gal1p and Gal3p proteins to 2% galactose from each of three conditions: glycerol-history, reinduction, and LTGR (Appendix Figs S5–S9; Materials and Methods). By binning cells with similar concentrations of Gal1p and Gal3p, regardless of when they reached those levels during the experiment, and measuring the concentrations for these cells 20 min later, we were able to estimate a vector that described the average change in Gal1p and Gal3p

concentration per unit time starting at that point on the state space (Appendix Fig S1). While the trajectories of individual cells may not be completely determined by Gal3p and Gal1p (Appendix Fig S2), we hypothesize that these two proteins are the most important factors controlling transient dynamics in this system and so these average vectors describe its dominant behavior.

The vector field unifies the three memory phenotypes into one consistent picture of cell behavior. At steady state, glycerol cultures express detectable Gal3p but no Gal1p, and glucose cultures express neither (Lohr *et al*, 1995). After 12 h of glucose repression, reinduction cultures contain evident Gal1p (Zacharioudakis *et al*, 2007) but little Gal3p (this study). The vector field representation illustrates how these three different initial transducer levels determine induction lag and population variability. Each of the three history media places the cells at different initial points in this state space (Fig 4). The appreciable presence of either transducer—Gal3p or Gal1p—is sufficient to drive the expression of the entire positive feedback loop, enabling these cells to ramp up induction quickly (Figs 1 and 2). However, cells that have been subjected to LTGR enter the galactose environment with no transducers to get the feedback loop going. They must wait for rare molecular interactions to produce a few molecules of Gal3p so the feedback loop can begin. We call this process *bootstrapping*, a colloquialism that refers to accomplishing a seemingly impossible task starting from the scarcest of resources (Evans, 1834). We suggest that this bootstrapping is the source of the long lag that has been observed following LTGR.

Single-cell dynamics on the vector field could also explain the differences in population-level induction patterns (i.e., unimodal vs. transiently bimodal) among the three history conditions (Biggar & Crabtree, 2001; Zacharioudakis *et al*, 2007) (Fig 1 and Appendix Fig S4). When only a few molecules of a protein are present in a cell, stochastic effects can dominate the molecular interactions involving that protein (McAdams & Arkin, 1997). LTGR-history cells are the only cases among our conditions where both initial transducer concentrations are low enough that we might expect stochastic induction behavior (Fig 4). We hypothesize that when cells are switched to galactose after LTGR, they must wait for infrequent, stochastic molecular events to activate the positive feedback loops. As a result, individual cells would wait widely varying times before starting induction. This would produce a slow, sticky region of the Gal1p/Gal3p state space where the concentrations of both proteins would be near zero and from which cells would slowly escape one by one while they bootstrap themselves into GAL network expression. When extrapolated to the population level, this dynamic would manifest as the slow, transiently bimodal induction pattern characteristic of LTGR where an initial distribution of uninduced cells shifts to one that is completely induced. In contrast, in reinduction and glycerol-history conditions, cells start with appreciable levels of at least one transducer, placing them outside the putative sticky region. The dynamics in this fast deterministic regime would result in a unimodal induction pattern at the population level.

Population measurements (Biggar & Crabtree, 2001; Zacharioudakis *et al*, 2007) cannot determine *why* the uninduced peak shrinks during the transiently bimodal period following LTGR. Does the uninduced fraction decrease because most of the cells in it activate the GAL network, thus switching to the induced fraction? Or does the fraction of uninduced cells shrink because a subpopulation of cells in it induces and starts to divide and demographically replace

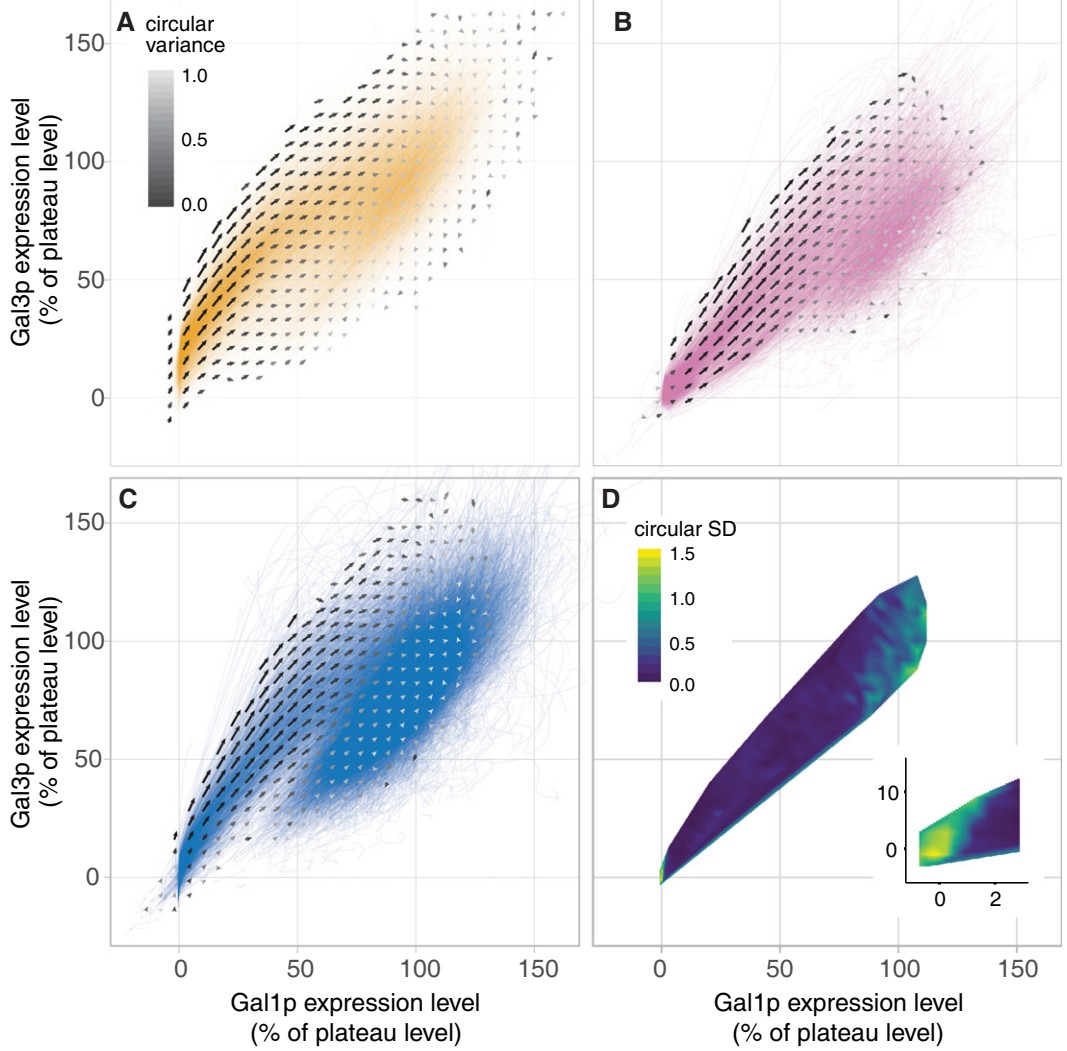

**Figure 3. Empirical vector fields depicting the flow through the Gal3p/Gal1p state space for the three history conditions.**

A–C  Vector fields for glycerol (A), reinduction (B), and long-term glucose repression conditions (C). Colored lines are traces of individual cells through the Gal3p/Gal1p state space and the darkness of the color at a point scales with the number of trajectories that pass through that point. Arrows represent the vector field. The state space is binned in steps of 6% in each direction, and an arrow summarizes the movement of cells in its bin from one time point to the next (Appendix Fig S1). The length of the arrow is half the mean velocity of cells, and the shade represents the circular variance (Pewsey *et al*, 2013) with black indicating consistency in the direction of displacement and white indicating inconsistency. The vector fields flow toward (100%, 100%).

D  Consistency of the vector directions during induction in the three experiments. The color indicates the estimated circular standard deviation of the mean vector directions for each of the three experiments (see Materials and Methods). The inset shows the region at the corner near (0%, 0%). Once cells leave this corner, the statistic drops to near zero indicating that cells are moving in a consistent direction regardless of experimental condition—they have lost their memories.

the rest? Our single-cell time courses clearly illustrate the latter process: The fully induced population is composed principally of the descendants of the earliest-inducing cells (Figs 1 and 5A; Movies EV1 and EV2).

The bootstrapping hypothesis makes a number of qualitative predictions for induction behavior. It suggests that cells starting in the region near (0%, 0%) will have long and variable lag times and that the variability in lag times explains the bimodality of LTGR-history induction: as individual cells escape from this sticky region, they leave the uninduced population to join the inducing subpopulation. Since the hallmarks of LTGR memory (length and variability of lag times) are consequences of the cells' tenure in the sticky region,

the bootstrapping hypothesis also predicts that once they accumulate appreciable levels of transducer and leave the sticky region, they should lose their memory of LTGR and their induction trajectories should match those of cells in reinduction and glycerol-history conditions. We tested these predictions and mapped the memory region using the empirical vector fields derived from our microfluidic experiments.

The results of our microfluidic experiments fulfilled the predictions of the bootstrapping hypothesis (Fig 1). Lag times after LTGR were much longer and more variable than after the other two history conditions (Fig 2; Appendix Fig S10; Appendix Table S2), and many cells failed to induce. As a result of the long and variable

    

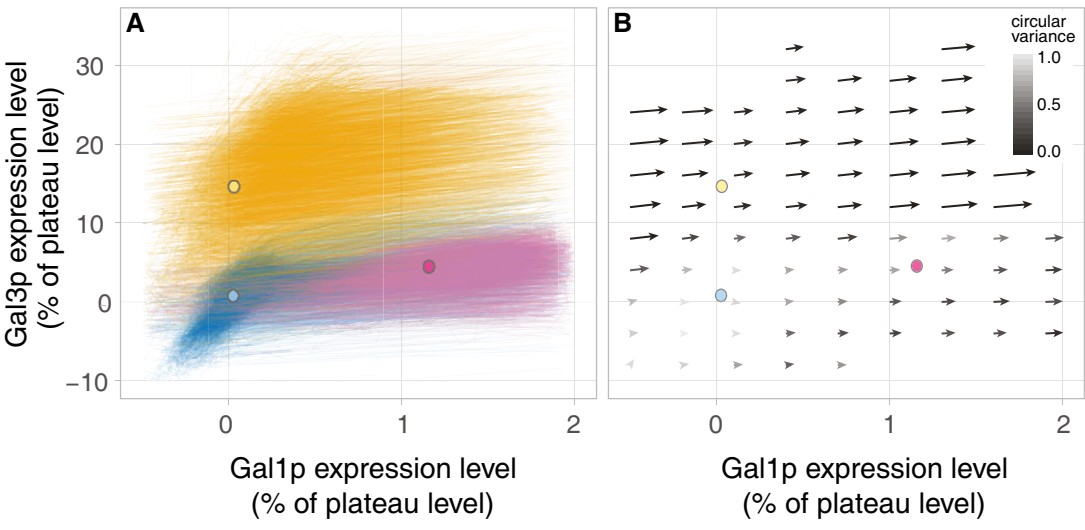

**Figure 4. Cell dynamics near the (0%, 0%) corner.**

A   Cell trajectories near the (0%, 0%) corner. Circles indicate the mean fluorescence values for each experiment at the time galactose was added.

B   The vector field near the (0%, 0%) corner. Circles are as in (A).

lag, a small minority of cells induced, grew, and dominated the population, while others remained quiescent or died, producing the transiently bimodal induction distribution that others (e.g. Zacharioudakis *et al*, 2007) have seen but not explained.

The influence of network induction dynamics on demography explains another difference between population profiles of GAL induction after LTGR and other nutrient histories. Following glycerol or reinduction conditions, the cell population induces uniformly, producing a single peak in the induction distribution that moves steadily from uninduced to induced (Figs 1, EV2 and EV3), producing a graded pattern of movement between peaks. By contrast, following LTGR, the peak representing uninduced cells does *not* shift; instead, it shrinks, while a different peak representing induced cells grows. This bimodal, non-graded pattern results from the fact that after LTGR, only a few cells at a time escape from the sticky region and traverse the middle region of the vector field to the fully induced peak. The rest of the population stays at the uninduced extreme of the field.

In the vector field illustrating cells from all three nutrient histories, the small region around (0%, 0%) was populated almost exclusively by LTGR cells, and their initial trajectories differed markedly from those in other conditions (Fig 4). Once induction was under-way, the dynamics were similar in all three histories (Fig 3D and Appendix Fig S11), which allowed us to map where the cells lose their memory of previous nutrient conditions (Figs 3D, 4, and EV3). Cells moved in a consistent direction in the Gal3p/Gal1p state space and transitioned rapidly from low Gal3p to plateau levels of Gal3p (Figs 1, 3, and EV3). Their trajectories slowed as they approached plateau levels of Gal3p, and the flow on the vector field curved toward the fixed point (100%, 100%). These dynamics suggest an unstable fixed point around (0%, 0%) and a stable fixed point around (100%, 100%). The trajectories of Fig 1 and the population distributions of Fig EV2 show variation in expression levels for both proteins after most of the population has induced. In all three conditions, over 80% of this variation can be explained by intercell

differences in average expression level, as opposed to fluctuation in a cell's measured expression over time (Appendix Table S3). This suggests that while the trajectories of inducing cells are generally similar, their final expression plateaus are cell-specific and influenced by other variables.

Although the results from the microfluidic experiments support the bootstrapping hypothesis, it is possible that other factors could also contribute to lag time between history conditions. For example, glucose represses the expression of a large proportion of the yeast genome, including the GAL genes, and is known to suppress GAL expression via several different mechanisms (Adams, 1972; Griggs & Johnston, 1991; Nehlin *et al*, 1991; Johnston & Carlson, 1992; Lamphier & Ptashne, 1992; Carlson, 1999). If glucose repression or its aftereffects linger in LTGR-history cells, this could cause the long lag we observe under those conditions.

Alternatively, resource constraints could contribute to induction delays. Gal1p is one of the most highly induced genes in yeast—it is upregulated 1,000-fold—which makes the GAL1 promoter a useful tool for genetic engineering (e.g., Fischer *et al*, 1988; Webster *et al*, 1988; Brand & Perrimon, 1993; Nevozhay *et al*, 2009) but also means that GAL network induction demands a large investment of resources (Johnston, 1987; Baumgartner *et al*, 2011). Polymerases, ribosomes, and carbon building blocks may need to be diverted from other genes, and a cell must have sufficient energy to build these proteins. Although cell populations with more energy reserves adapt as a whole more quickly to galactose (Reiner & Spiegelman, 1947; Spiegelman *et al*, 1947), cells do not use this "battery" power to propagate. Instead, when cells are switched from long-term glucose to galactose, every cell in the population abruptly stops growing or dividing and remains in stasis for hours (Fig 5B; Movie EV2). Although they are drowning in galactose, they have no GAL proteins and are unable to use it. Fueling initial GAL induction via stored energy/carbon reserves alone may be a formidable effort that only a few cells are able to muster (Fig 5A). Only the cells that eventually begin GAL induction ever resume growth in our

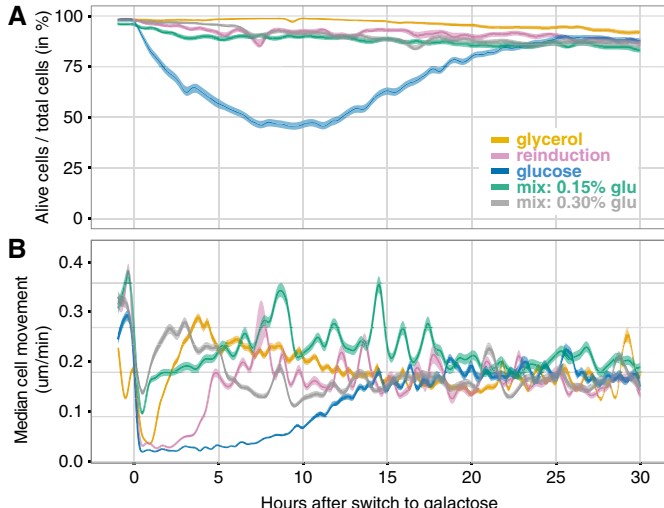

**Figure 5.  The switch to galactose imposes a heavy cost after long-term glucose repression (LTGR).**

A    An estimate of cell viability after the switch to galactose. Cells were classified as alive or dead (Materials and Methods; Appendix Fig S8). Dying cells were classified as alive, so these curves represent upper bounds for the fraction of viable cells in the populations. Lighter bands indicate 95% confidence intervals for the proportion (the darker lines).

B    Average cell movement in microns per minute for cells in a field of view as estimated measuring physical displacements of individually tracked cells in bright-field images taken every 2 min. Median cell movement is a surrogate for the amount of cell division since cells in the microfluidic device are confluent and push each other when they divide. After LTGR, the switch to galactose is accompanied by a 6-h-long pause in cell movement as cells bootstrap themselves into GAL network induction followed by a subsequent slow 7-h recovery as the first cells to induce and their progeny take over the population (Movie EV2). Lighter bands indicate 95% confidence intervals for the median (the darker lines) estimated by bootstrapping.

observations (Movie EV1). They rapidly outcompete the rest of the still-starving population, dooming them to demographic oblivion. The stately population-level picture of a bimodal population resolving into a unimodal, fully induced one is, in fact, a process of lineage selection: some cells never manage to induce before they die, and others only do so too late. By contrast, the cultures that start with enough Gal3p or Gal1p to begin GAL induction quickly—and thus use galactose to fuel further GAL induction—pause only briefly before growth resumes in the new medium (Fig 5B), and the fully induced population preserves many of the original cell lineages (Fig 1; Movies EV1 and EV2).

If our resource-constraint hypothesis is correct, then adding a small amount of glucose to the galactose media should give cells a boost—extra resources to build their first GAL proteins (Spiegelman *et al*, 1947). On the other hand, if lingering glucose repression were a cause of the induction delay, then glucose in the induction medium should prolong it. We found that adding 0.15 or 0.30% glucose speeds up GAL network induction (Fig 6 and Appendix Fig S12): In both mixed-sugar conditions, half the inducing cells reach 10% of their plateau Gal3p expression levels within 2.1 h (Fig 6B), while LTGR cells take 6.4 h to reach the same point (Fig 2).

These results rule out lingering glucose repression as an explanation for the long lag times we observe after LTGR for times beyond 2.1 h. They also point to energy or other resources as important limiting factors for the successful transition to metabolizing galactose. While the bootstrapping process imposes a lag on GAL activation when cells have no initial transducers, this process speeds up if cells can draw upon energy and carbon from the induction media (Fig 6B). Without usable external energy, cells must fuel induction using only stored energy reserves and balance this against the need to live off these reserves in the meantime. Once GAL network induction is underway, galactose-derived resources become available to help fuel further induction (Appendix Fig S13). Having other sources of energy available also gives cells the option whether to induce the GAL network at all (Escalante-Chong *et al*, 2015).

Resource constraints alone, however, cannot explain the length and variability of the delay following LTGR. If they could, then the timing and variability of GAL network induction after long-term and short-term (reinduction) glucose repression should be the same: In both cases, cells would have had at least 12 h in rich glucose medium to build up their reserves. Or, if 12 h of exposure to glucose were not enough, reinducing cells should take even longer to induce, and lag length should be even more variable. In fact, induction after LTGR takes nearly three times longer than reinduction and is twice as variable (Fig 2 and Appendix Table S1), so bootstrapping from initially absent transducers must play a role in lag times and variability. Conversely, if resource constraints played no role, then giving the cells extra resources during the critical early-induction period would have no effect. Instead, induction was faster when we supplemented cells with glucose. Therefore, both bootstrapping and resource constraints affect the lag.

Although the *process* of bootstrapping after LTGR may be stochastic—because of the small numbers of molecules involved—it is entirely possible that *which* cells are successful, or perhaps the pool of cells that are able to induce at all, is influenced by aspects of cell state that we did not measure. When induction is easy and fast because of initially present transducers, there is no starvation period, and almost any cell can mount the effort required for induction. When induction is slow and requires surviving starvation, fewer cells have the energy or carbon reserves to last until they can bootstrap themselves to the induction level where they can eat galactose. Resource levels may also speed the bootstrapping process itself: When we supplemented cells with glucose after LTGR (Fig 6), they induced more quickly.

## Discussion

In a changing environment, cells must constantly adjust their gene expression to take advantage of the energy sources currently available to them. Failing to respond properly can be fatal. Most recent research on how and when yeast cells induce the GAL genes has focused on how information flows through the regulatory network. Here, we provide support for an idea first advanced nearly 70 years ago (Spiegelman *et al*, 1947) but neglected in recent explorations of this system: that the availability of cellular resources is an important additional constraint on GAL induction.

In addition, we offer a unifying explanation of previously documented cellular memory phenotypes. Others have shown that yeast

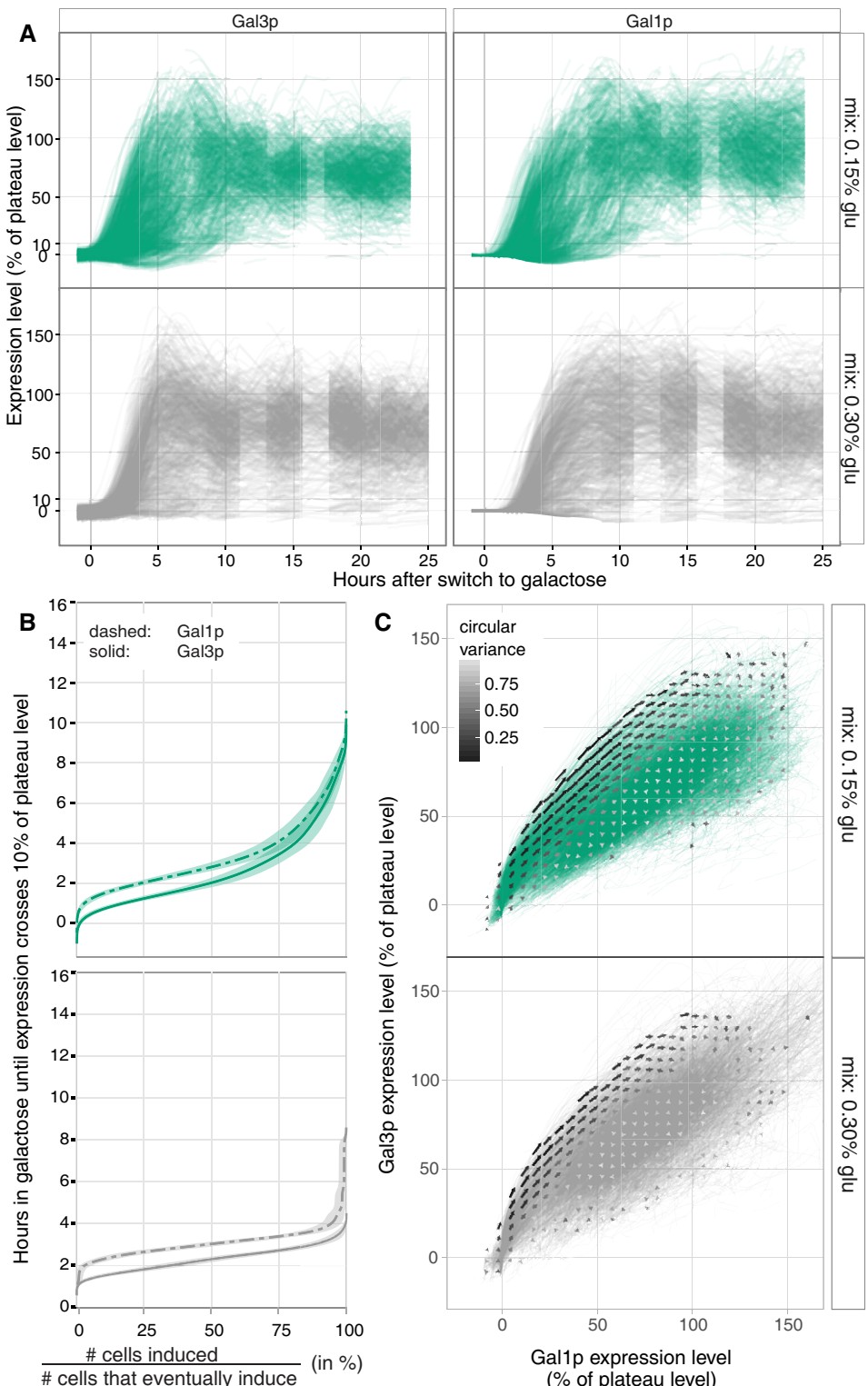

**Figure 6.  Adding a small amount of glucose speeds GAL network induction.**

A   Single-cell time courses when cells are moved from long-term glucose repression to a mix of 0.15% glucose/2% galactose (green) or 0.3% glucose/2% galactose (grey). For both Gal3p and Gal1p, cells initially overshoot their long-term plateau levels.

B   Empirical cumulative distributions as in Fig 2 with 95% confidence interval bands. Additional glucose shortens the time until induction, but there is still appreciable between-cell variation with 0.15% glucose.

C   Vector fields as in Fig 3A–C. The vector fields noticeably overshoot then curve back around toward (100%, 100%). The flows on the Gal3p/Gal1p state space are similar in these mixes and otherwise similar to the conditions in Fig 3.

populations can induce uniformly or bimodally, quickly or slowly, depending on their previous nutrient conditions. We tested whether the Gal1p and Gal3p levels in cells emerging from each of three nutrient histories could explain the differences among the induction patterns, particularly the long delay and bimodal distribution following LTGR. By measuring Gal1p and Gal3p expression levels in thousands of individual cells and using these to create a living version of a mathematical construct—a vector field—we established that the need to bootstrap the Gal1p/Gal3p positive feedback loop from undetectable levels of both transducers contributes to the long lag times and induction variability that characterize the LTGR memory phenotype. By tracking gene expression in individual cells over many hours, we distinguished between growth and induction effects, demonstrating that the minority of cells that manage to achieve GAL induction after LTGR using stored resources come to dominate the population while other cell lineages are lost.

Activating the GAL network from a repressed state with no initial transducers requires resources that most cells do not muster and depends on chance molecular interactions that most cells do not experience. The minority of cell lineages that overcome these barriers take over the population. The living vector fields synthesize results from five different nutrient conditions to reveal that GAL induction behavior depends on whether cells need to bootstrap the positive feedback loops in the network from initially absent transducers and that the bootstrapping process is constrained by cellular resources.

# Materials and Methods

### Yeast strains

The two yeast strains used in this paper were derived from BY4741 (Baker Brachmann *et al*, 1998) (*his3Δ, leu2Δ, met15Δ, ura3Δ, MATa*), which is a derivative of S288C. YSR0145 had a Gal1p C-terminal fusion to yECerulean and a Gal3p C-terminal fusion to 2x-yECitrine.

To make YSR0145, we replaced the GAL1 stop codon with yECerulean-ADH1t-pTEF-SpHIS5-TEFt from pKT101a (from Natalie Cookson) which is identical to pKT101 (Sheff & Thorn, 2004) except with CFP replaced by yECerulean. We used the oligos CGTCTCTAAA CCAGCATTGGGCAGCTGTCTATATGAATTAggtgacggtgctggttta and GTTATTATTGCGTATTTTGTGATGCTAAAGTTATGAGTAGtcgatgaa ttcgagctcg to insert this translational reporter into the yeast genome (the upper case pairs with the yeast genome and the lower case pairs with the plasmid). We then made a translational reporter for Gal3p by inserting 2x-yECitrine from plasmid pSRS001 (Appendix Fig S3) before the stop codon of GAL3 using the oligos AGTTTCGAAGCCTGCCTTGGGTACTTGTTTGTACGAACAAtacgctgca ggtcgacggat and CTTTTAATATTTAAAGGTTGTTCCAAGAAGGTGT TTAGTGttataattggccagtctttt. 2x-yECitrine consists of two copies of yECitrine connected by a linker sequence, which approximately doubles the fluorescence intensity of a single yECitrine reporter and allows detection of Gal3p even at low expression levels.

As an internal control, we constructed YSR0140, an isogenic strain to YSR0145 without Gal reporter proteins but with an mCherry (Shaner *et al*, 2004) nuclear marker to enable us to distinguish these cells from YSR0145. We constructed strain YSR0140 by replacing the stop codon of the nuclear-localized NRD1 in BY4741 with mCherry-pTEF-NAT-TEFt (Shaner *et al*, 2004) using the oligos GAATATGCTTAACCAACAGCAGCAGCAACAACAACAAAG Catggtgagcaagggcgagga and TTTTATGTACTATGAGCAAATAAAGGG TGGAGTAAAGATCatcgatgaattcgagctcg with plasmid pBS35+25. This plasmid is identical to plasmid pBS35 (https://www.addgene.org/ 83797/) except with hph replaced by NAT.

### Growth conditions and media

Cells were grown at 30°C in low-fluorescence synthetic dropout media with complete amino acids and the appropriate carbon source. For microfluidic experiments, cells were cultured in *history medium* (2% glucose for LTGR, 2% galactose for STGR, or 3% glycerol) for at least 10 generations or 24 h, whichever was longer. For the final 10 generations, cell density was kept at $OD_{600} < 0.15$ to maintain effectively constant sugar concentrations. Following this, cells were loaded into the microfluidic device according to a published protocol (Ferry *et al*, 2011) and grown in history medium until approximately confluent. We aimed for the majority of cells to be experimental cells with enough control cells to be able to estimate background fluorescence well. The microchemostat media was then switched to an induction medium as described in the text and imaged for $\geq 25$ h. For reinduction experiments, the microfluidic medium was instead switched to 12 h 2% glucose then $\geq 25$ h 2% galactose. The media took less than 1 min to completely switch in the cell chamber.

The cells were imaged using a Nikon TiE-inverted microscope with the Nikon Perfect Focus System and a 60× oil immersion objective (CFI Plan Apo VC 60× Oil, 1.4NA, 0.13WD). A Prior Lumen 200 with a mercury lamp was used with a Sutter Lambda SC shutter and three filter cubes for fluorescence imaging of YFP (Semrock YFP-2427B), CFP (Semrock CFP-2432C), and mCherry (Nikon Y-2E/C). The light source was limited to 25% maximum intensity. Bright-field images were taken with a Nikon halogen lamp with exposure controlled by a Sutter shutter. 16-bit greyscale images were captured using a Princeton Instruments PIXIS1024b cooled-CCD camera with a 1,024 × 1,024 pixel, (1.3 × 1.3 cm) chip. The temperature on the microscope stage during imaging was maintained at 30°C with a Nevtek ASI400 air stream incubator and a custom-rigged microscope enclosure.

We imaged cells briefly and infrequently to minimize photostress and photobleaching (Appendix Fig S4): bright-field 50 ms every 2 min; mCherry 50 ms every 40 min; 2x-yECitrine and yECerulean for 600 and 200 ms, respectively, every 20 min. Automated time-lapse imaging was controlled by a PC running custom routines in µManager (Edelstein *et al*, 2014) and MATLAB (The MathWorks, 2014). In order to facilitate the image processing, we also took a series of 50 ms bright-field images above and below the target focal plane. Because Gal1p-yECerulean expression varies over a huge range, a single exposure time would not capture the entire dynamic range without saturating. To circumvent this, we took a series of four successive 50 ms exposures and then summed them to get a total yECerulean image that could accommodate the yECerulean dynamic range. We used a similar method for Gal3p-2x-yECitrine taking a series of twelve 50 ms exposures.

## Cell segmentation

We developed a MATLAB-based (The MathWorks, 2014) cell segmentation and tracking pipeline based on a published pipeline (Ricicova *et al*, 2013) to extract data from the images (Appendix Figs S5–S9).

The cell segmentation pipeline has a number of free parameters, some of which are more influential than others. It was common for a parameter set to work for most but not all of the cells in an image and for another parameter set to pick up the missed ones. We wrote a MATLAB GUI to help pick sets of parameters that would collectively identify and segment the cells in a dataset and used three parameter sets for each dataset. We segmented the cells based on each parameter set and then merged them to make a final cell segmentation (Appendix Fig S5).

We took a series of images above and below the focal plane and then max-merged subsets of these to make above-plane and below-plane images. When we subtracted the below from the above images, constructive interference from the diffraction patterns around the cells roughly outlined the cell boundaries (Ricicova *et al*, 2013). We then performed a series of image processing steps on this image. We (i) thresholded this image using Otsu's method (Otsu, 1979), (ii) removed small objects, (iii) morphologically closed the image to fill holes, identified the insides of cells through a combination of (iv) dilation and erosion to identify and remove cell membranes and (v) inverting the segmentation mask after (iii), (vi) smoothed the edges of the objects, (vii) separated clustered cells by the watershed method, (viii) filled holes, (ix) removed objects that were too small, too large, or touched the edges, and (x) ensured that all segmented objects were 8-separated. Vacuoles not infrequently interfered with proper segmentation, leaving us with objects that looked like Cs or doughnuts. We wrote a script to assess these segmented objects based on area and statistics relating to circularity, merge them into a single cell if appropriate, or fill them in otherwise. This gave us three cell segmentation masks for a frame. We merged them using a pipeline similar our tracking pipeline described below that matches cells between frames.

## Cell tracking

Once a segmentation mask was created for each frame, we tracked cells across frames. This involves determining how segmented objects in one frame correspond to the segmented objects in the next. The tracking algorithm of Ricicova *et al* (2013) matches cells across frames using the Hungarian algorithm (Kuhn, 1955) to minimize the distances between the centroids of segmented objects in one frame and the predicted locations of objects from the previous frame based on their previous location and velocity. Unlike Ricicova *et al*, our interval between frames was only 2 min, and so there was limited movement between frames. For our images, their assignment procedure mismatched a large fraction of cells. However, because the cells did not move much over the 2 min, we were able to rely primarily on overlap between segmented objects to track cells between frames. We devised a pipeline (Appendix Figs S6 and S7) to make these assignments and to cope with both cell movement and improper segmentation due to splitting a single cell into two objects, combining two cells into one object, and filling in a missing cell in one or two frames.

When confluent, approximately 2,000–3,000 cells were present in a field of view in the microfluidic device. We were able to reliably track a large fraction of cells across hours and, crucially, measure many cells entirely through their induction of the GAL network.

## Classification of cells as alive or dead

Dead cells autofluoresce for a time but also take on a characteristic appearance under bright-field illumination (Appendix Fig S8). We trained a random forest classifier (Breiman, 2001) to distinguish between alive and dead cells based on the bright-field images and classified all segmented objects as alive or dead. We also used this classification to improve the cell tracking by splitting "cells" that were classified as dead for a number of frames but then were resurrected.

## Estimation of protein concentration

We estimated protein concentrations within the cells based on Gal1p-yECerulean or Gal3p-2x-yECitrine fluorescence intensities. To measure these, we first added the separate 16-bit images to make a composite image where the range could be larger than $2^{16}$. To remove the effects of variable background signal, we calculated the median background fluorescence level for non-cell areas of each image and subtracted that from the image. This gave all images a comparable baseline. During the experiments, some cells developed puncta which showed up as small, high-intensity fluorescent spots, particularly in the 2x-yECitrine channel. To prevent these from distorting the fluorescence intensity concentrations, we removed them in a two-step process. After scaling the intensities of a fluorescence image to be between 0 and 1, we identified cells where the skewness of the distribution of pixel intensities in a cell was > 1.5 and then removed any pixels in that cell with intensity values greater than seven median absolute deviations above the median. This procedure reliably identified pixels in puncta, particularly in cells with low expression where the excess fluorescence would have the most effect. The mean of the remaining pixel values gave a much better estimate of the protein concentration in the cell.

We distinguished control from experimental cells using the nuclear-localized mCherry marker. We calculated several statistics on the magnitude and spatial organization of the mCherry signal as well as the yECerulean signal in each cell and used a custom MATLAB GUI to train a random forest (Breiman, 2001) to distinguish control cells from experimental cells from bad cells (i.e., cells that had likely died and had intense autofluorescence in all channels). We used a cost function that was conservative in classifying a cell as control so that our estimated normalization factors would not be skewed high. The false identification rates based on out-of-bag estimates were near zero for control cells and at most a few percent for experimental cells—not high enough to affect our results. We used the cell type classifications to do a final curation of the cell tracking by separating cells that had been classified as experimental in some frames and control in others into two different cells. If a cell were marked as bad in a frame, it was classified as bad for all subsequent frames (Appendix Fig S9).

Based on the cell masks and the background-subtracted fluorescent images, we measured total pixel intensity within a cell and the area of the cell and from these calculated the average pixel intensity, which we took as proportional to the protein concentration in the cell and used as the cell's fluorescence level. Because the control cells had neither 2x-yECitrine nor yECerulean, we could use them as an internal control for autofluorescence and any bleedover in fluorescence due to out-of-focus light from the experimental cells. To remove any image artifacts that were segmented and tracked as cells, we restricted the dataset to cells that were present for at least five bright-field frames. We also ensured that there were at least 50 control cells per frame. For each frame, we estimated the median fluorescence level across the control cells and subtracted this from all the cells in the frame. In that way, zero for each frame was set to the median control cell fluorescence level.

### Trajectory smoothing for fluorescence and scaling each experiment to be between 0 and 100% expression

To remove high frequency noise from the individual cell fluorescence trajectories, we smoothed them by first convolving them with a Gaussian kernel and then smoothing the result using loess regression with a short span. Upon dying, fully induced cells decline in expression. Our classification of cells as alive or dead (above) allowed us to identify most of these cases and remove the drops from the analysis. To catch any others, we also eliminated parts of a cell's trajectory after it reached its peak where expression persistently (80% of postpeak frames) dropped in expression in either fluorescence channel (indicating possible photobleaching or other problem) or dropped monotonically over five frames by 50% as there were likely to represent tracking errors or cell dying. To normalize across experiments, we scaled the Gal3p-2x-yECitrine and Gal1p-yECerulean data between uninduced (0% expression) and fully induced (100% expression). About 0% expression was the control cell median baseline described above. For 100% expression, we estimated where the fluorescence level plateaued. After examining the fluorescence trajectories for each experiment, we identified a period of time where the cells appeared to plateau, took the maximum fluorescence intensity for each cell in that time period, and took the median of these maxima to be our estimate of full induction.

### Vector field estimation

To estimate the vector field for an experiment, we restricted our focus to between 0 (galactose exposure) and 900 min. This is well after the time for GAL network induction in all experiments. We divided the Gal3p-Gal1p state space into bins of 6% plateau expression on each side. Within each bin, we identified all cells with Gal3p and Gal1p expression levels in that bin which were also tracked into the next fluorescent frame. Data from these two time points gave us a velocity for each cell. The mean velocity within the bins is represented in the vector field plot by the length of the arrows (scaled by 1/2 so that the arrows do not overlap). The lightness of the arrows indicates the circular variance of the vector directions, a statistic that ranges from 0 (low variation in directions) to 1 (high variation) (Mardia, 1972) (Appendix Fig S1).

To estimate the consistency of the vector fields during induction, we filtered the data so as to remove much of the plateau period. For each experiment, we determined the frame at which 95% of the inducing cells had induced (Fig 2). For each cell, we assigned a cutoff frame as follows: for cell that spent at least five frames above 85% plateau level for Gal1p before the experiment-specific frame determined above, the cutoff frame was the fifth frame; for all other cells, the cutoff frame was the experiment-specific frame. Only data at or before this cutoff frame were used in the consistency calculations.

The consistency statistic is an estimate of the average circular standard deviation of mean vector directions in each of the experiments LTGR, reinduction, and glycerol. We divided the data into bins and only used bins for which there were at least 10 cells in each of the experiments. For each experiment, we sampled the data 500 times with replacement with a sample size of 10 and calculated the mean vector direction for each replicate. For each of the 500 replicates, we took one of the resampled mean vector directions for each experiment and calculated their standard deviation. Our statistic is the mean of these 500 circular standard deviations. In Fig 3D, we estimated the statistic at a fine spacing by linear interpolation.

### Variance decomposition

To estimate the fraction of cells' "steady-state" variation that was due to consistent intercell differences, we subtracted the mean cell fluorescence for each time point between 15 and 25 h after galactose exposure (when the cells were at their plateau level), then fit linear mixed-models to the fluorescence levels, with experiment as a fixed effect and cell ID as a random effect. We used the $R^2_{GLMM(c)}$ statistic (Nakagawa & Schielzeth, 2013; Lefcheck, 2016) to quantify the fraction of variation explained by cell-specific differences and calculated 95% confidence intervals by bootstrapping across cells.

### Software

Data processing and analysis were done using custom scripts in MATLAB (The MathWorks, 2014), MySQL (MySQL; www.mysql.org), and R (Wickham, 2011; Bates *et al*, 2015; RStudio Team, 2015; Wickham & Francois, 2015; R Core Team, 2016; Tsagris & Athineou, 2016), and all figures were made using ggplot2 (Wickham, 2009) or MATLAB (The MathWorks, 2014).

### Data availability

The data and analysis scripts can be found in the UCSD Digital Collections under the reference: Stockwell, Sarah R; Rifkin, Scott A (2017). Data from: A living vector field reveals constraints on galactose network induction in yeast. *UC San Diego Library Digital Collections.* http://doi.org/10.6075/J0C24TCX.

**Expanded View** for this article is available online.

## Acknowledgements

This work was supported by a San Diego IRACDA postdoctoral fellowship to SRS (NIH K12 GM068524), a Human Frontiers Science Program Young Investigator award to SAR (HFSP RGY0073/2010), and the San Diego Center for

Systems Biology (NIH P50 GM085764). We thank Christian Landry, Mike Ferry, Ivan Razinkov, Jeff Hasty, Dan Pollard, Sharon Tracy, Sidney Kuo, and other members of the Rifkin lab and UCSD Microscopy and Modeling group for technical help, discussion, and suggestions.

## Author contributions

Both SRS and SAR conceived of the study, designed the experiments, analyzed the data, and wrote the paper. SRS carried out the experiments. SAR wrote the software for image and data analysis.

## Conflict of interest

The authors declare that they have no conflict of interest.

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
