## [Review Process File · Molecular Systems Biology]

A living vector field reveals constraints on galactose network induction in yeast

Sarah Stockwell and Scott Rifkin

Corresponding author: Scott Rifkin, University of California, San Diego

Review timeline:

Submission date:	14 September 2016
Editorial Decision:	27 October 2016
Revision received:	30 November 2016
Editorial Decision:	19 December 2016
Revision received:	20 December 2016
Accepted:	21 December 2016

Editor: Maria Polychronidou

Transaction Report:

1st Editorial Decision

27 October 2016

Thank you again for submitting your work to Molecular Systems Biology. We have now heard back from the two referees who agreed to evaluate your study. As you will see below, the reviewers appreciate the interesting findings presented in the study. However, they list several issues, which we would ask you to address in a revision.

The reviewers' recommendations are rather clear so there is no need to repeat the points listed below. Both reviewers mention that adding modelling analyses would significantly enhance the impact of the study. Please let me know in case you would like to further discuss any specific point raised by the referees.

REFeree REPORTS

Reviewer #1:

A. Summary of this work

The galactose network in budding yeast is a model network for understanding how cells make decisions because its genes turn off and on in a switch-like (bistable) fashion. Although this system has been studied at the systems-level over the past decades, including properties such as hysteresis (i.e., cell's past exposure to sugars affects the on / off behaviour of the galactose genes), it still has many properties that remain poorly understood. Stockwell and Rifkin address one of these questions: How does a long term repression by glucose affect the induction dynamics of the galactose network? They answer this question by measuring a "flow diagram" - a concept from dynamical systems theory in which each point in a "phase space" has a "velocity arrow" that dictates

how fast and in which direction the cell moves next. Here, the phase space is a two dimensional plane spanned by the abundance of two upstream transcription factors, Gal1 and Gal3, that control the GAL network as a complex function of the type and concentration of extracellular sugars. Gal1 and Gal3 proteins were fused to spectrally orthogonal fluorescent proteins so that the authors could simultaneously measure their levels in individual cells on a microscope. By tracking individual cells' Gal1 and Gal3 protein levels over many hours, the authors showed that yeasts that were exposed to glucose for a long time had nearly zero levels Gal1 and Gal3 proteins. Since at least one of Gal1 or Gal3 levels have to go above a certain threshold to turn on the Gal network, the authors' flow diagram found that the cells that were growing in glucose, which represses the Gal network, for a long time took a longer time to "climb up" (a.k.a. "bootstrap up") and go over the activation threshold. Importantly, the authors show that many of the cells pay a heavy fitness cost when they are suddenly introduced to galactose after having been in glucose for a long time (Fig. 3), including some cells dying as a result and only the ones that, by chance, induce their GAL network earlier than their kins, can survive. Moreover, the authors show that they can help these cells more "gradually" cope with the sudden introduction of extracellular galactose by giving the cells a helping hand: a small (non-repressing) amount of glucose so that the cells eat to fuel their initial climb towards the activation threshold for turning on the GAL network.

B. Overall evaluation

Despite its apparent simplicity, this manuscript presents conceptually and biologically important findings. I think the most conceptually interesting aspect is the authors' measurement of the vector fields ("living fields") (Fig. 2) which serve as a novel tool to visualize gene expression dynamics. Such vector fields have only been extensively measured for a few networks in a few cell types (e.g., excitable networks in *B. Subtilis* competence network). I think the authors present one of the more extensively and carefully measured flow field among the examples that I have seen. The demonstration that such a flow field can be measured is relevant to a broad community of biologists, beyond those interested in the GAL network and micro-organisms. For example, the authors' study is conceptually relevant to stem cell biologists who are interested in measuring the "Waddington landscape" for distinct cell fates using single-cell RNA-seq. The authors have essentially measured the Waddington landscape for the galactose network in yeast. This work is an important contribution to systems biology.

The authors' main biological finding, which also reaches beyond the GAL network and micro-organisms, is a strategy for cells to cope with a sudden change in the extracellular nutrient conditions. The authors show that a prolonged repression of the GAL network by a long term growth in glucose depletes the Gal1 and Gal3 levels to nearly zero and thus the yeast requires a longer time for the positive feedback to ramp itself up and pass the activation threshold for switching on the GAL network. Fig. 4 captures this finding in a striking way: Yeasts need help by being given a small amount of glucose so that they can use that glucose to initially fuel their powering up of the positive feedback loop. I find that this is a beautiful result that is likely relevant to other systems and cell types because there are many scenarios in which microbes and mammalian cells need to switch their gene expression programs from one form to another due to a sudden changes in their environment. In a sense, the authors' discovery strongly suggests that a sudden change in the environment that is supplemented with a small bit of the previous environment (in this case, small concentration of glucose) makes a difference in the fitness of the cells.

Finally, this manuscript is very clearly and interestingly written. The authors explain a complex concept in a way that should be accessible to a wide range of readers.

Given these strengths, I enthusiastically support this manuscript for publication if the authors can address some (not necessarily all) of my concerns below with their current data or mathematical models. I do not think any new experiments are necessary for the current scope of the work.

C. Major points

C1. Density of vector fields: The colors in Fig. 2A, B, C, E represent the "circular variance". But perhaps the more important feature is how many of the single-cell trajectories pass through each pixel in the Gal1-Gal3 phase plane. It would be good to plot the vector fields with the density of single-cell trajectories. The circular variance only gives the variability in the direction and magnitude of the vector from one cell to another in a given pixel, but it does not give the probability that a cell would be found in that pixel. I think the density is an important feature of the vector field.

C2. Depth of the "sticky point" and mathematical model: For me, the major weakness of the

manuscript was that there was no mathematical model despite the richness of their data in a well characterized system that would make modelling a possibility. The main claim of the authors is that the vector field (Fig. 2) reveals that cells that underwent a Long-Term Glucose Repression (LTGR) are stuck in a "sticky point" that's near the $\text{Gal3p} = 0$ and $\text{Gal1p} = 0$. But can this qualitative statement be made stronger with a quantitative model? How "sticky" is this place? Perhaps the authors can write down the differential equations for the galactose network (as in Acar et al, Nature 2005) and then approximate the stochastic gene expression using Fokker-Planck approximation. From this Fokker-Planck approximation, one can calculate how long it would take for a cell to escape the "potential well" that might characterize the "sticky location" ($\text{Gal3p} \sim 0$, $\text{Gal1p} \sim 0$). Admittedly, this is easier said than done and perhaps it's beyond the scope of the current study. But I think some estimation of the depth of the well that traps the cells in the "sticky region" (e.g., by linking the "hours in galactose until expression crosses 10% of plateau level" in Fig. 4B with a model that estimates this depth) would make the claim more quantitative. I don't expect an extensive model but some quantitative reasoning would strengthen the work.

D. Minor points

D1. Stable / unstable fixed points in the vector field: Indicate where the stable and unstable fixed points are (i.e., fixed point values for Gal3p and Gal10p) in the vector fields (Fig. 2).

D2. Some of the labels and vector arrows are too small to see on paper or regular sized PDF: Examples include the vector arrows in Fig. 2 and Fig. S12, the labels for hours and sugar types in Fig. S2, and Fig. S10.

D3. Some minor typos here and there.

Overall, I enjoyed reading this manuscript. I enthusiastically support this work for publication particularly due to its conceptual novelty and the revelation of a seemingly general strategy for cells to cope with a sudden change in the nutrient environment.

Hyun Youk.

Reviewer #2:

This manuscript aims to investigate how yeast cells respond to changing the growth media from glycerol or glucose to galactose. A new form of representation, called "living vector field" is proposed, which consists of temporal changes in the levels of two proteins plotted against each other. This representation provides a two-dimensional visual representation of protein level dynamics, and should reveal the cellular memory of earlier environments in which the cells grew. Such living vector fields constructed for two key activator proteins, GAL1 and GAL3 reveal that they are so strongly depleted after long-term glucose exposure that the cells induce with very long and variable delays. This results from a bimodal induction profile that originates from lineage selection, where a subpopulation of cells that manages to express the GAL1 and GAL3 activators outcompetes the cells that cannot. Moreover, and intriguingly, it is found that cells that fail to express the activators die during induction. This is a striking and very important observation that sheds light on a long-standing but crucially important dichotomy: when cells induce the galactose network, is the population-level response based on gene regulation or phenotypic selection? Answering this question is crucially important because it can change fundamentally the interpretation of microarray data of cells exposed to changing sugar conditions and beyond. Indeed, the same mechanism is likely to occur when cells respond to various stresses. This means that these observations generalize not only to other environmental changes, but also other cell types. Based on these considerations, I believe that this manuscript fills a very important gap in the field and I strongly support its publication in *Molecular Systems Biology* once the following comments are addressed.

(1) Throughout the manuscript, GAL1 and GAL3 are called "inducers". However, the standard use of the term "inducer" refers to a small molecule that causes gene expression increase by binding a transcription factor. To avoid this confusion, Gal1p and Gal3p should not be called "inducers". In fact, the sugar galactose is the inducer for the GAL network. Gal1p and Gal3p could perhaps be

called "positive regulators".

(2) Font size is too small (almost invisible) in Fig. S2 and other figures. Please check font size throughout the manuscript and the SI when this comment is applicable.

(3) The concept of the vector field seems to be useful. It would be helpful to illustrate with a cartoon how the field arrows are generated. It would also be useful to be very explicit in describing what the vector field reveals that other visualization approaches could not. Likewise for the variance of the vector field.

(4) One critical piece of information missing from the vector field plots is time. Would there be a way to indicate time by some faint contour lines that the vectors are perpendicular to? I think this would be useful.

(5) What does really happen after LTGR? I would guess that the protein Gal2p must play a key role because it takes up galactose. Can the cells ever induce if GAL2 is deleted? The role of Gal2p should be clarified in a couple of sentences. Maybe the depletion of Gal2p is equally important as the depletion of Gal1p and Gal3p.

(6) A mathematical model capturing the observations (even if simple) would be very useful for the paper. Perhaps there is no need to go into full-detail modeling of the GAL network - a simple model including the activators and their degradation, as well as their effect of cell division or death rates may be sufficient. This would also increase the interdisciplinary relevance of the manuscript.

(7) The term "bootstrapping" is heavily used, but not clearly explained. It is a contracted idiom that may not be familiar to many non-native speakers. Therefore, the Authors should clarify what is the general meaning of bootstrapping in everyday English, and how does the cells' behavior relate to this meaning of the word "bootstrapping"?

(8) The current manuscript lacks a Methods section in the main text. It may be worth considering to move some of the methods to the main text. A Discussion would also be useful.

(9) The set of image processing Matlab scripts with an example image series should be made available online as Supplementary Material.

(10) After mentioning that "Gal1p is one of the most highly induced genes in yeast - it is upregulated 1000-fold - which makes the GAL1 promoter a useful tool for genetic engineering" it may be useful to cite a couple of references, for example, PMID:27111147 or PMID:19279212. Other useful references may be PMID:24453942 and PMID:25626086.

1st Revision - authors' response

30 November 2016

Response to Reviewers

MSB-16-7323 "A living vector field reveals constraints on galactose network induction in yeast"
Sarah R. Stockwell and Scott A. Rifkin

We appreciate the editor's and reviewers' comments and suggestions and have changed the text and figures where appropriate to address them. Our detailed responses are below.

#1. Organization of the paper

We have reorganized the paper by moving Materials and Methods to the main text, including a Discussion, moving tables and most supplemental figures to an Appendix, and moving 3 additional figures (including a new one) and the movies to Expanded View.

#2. Reviewer 1. Density of vector fields: The colors in Fig. 2A, B, C, E represent the "circular variance". But perhaps the more important feature is how many of the single-cell trajectories pass through each pixel in the Gal1-Gal3 phase plane. It would be good to plot the vector fields with the

density of single-cell trajectories. The circular variance only gives the variability in the direction and magnitude of the vector from one cell to another in a given pixel, but it does not give the probability that a cell would be found in that pixel. I think the density is an important feature of the vector field.

While the greyscale in the vector field figures indicate the circular variance, the shade of the colors indicates the density, as the reviewer suggests. The individual cell trajectories are actually all represented in this figure, and they are each translucent. The effect of this is that the more trajectories that cross a given pixel, the more opaque that pixel is. This presentation, however, suffers from three limitations. First, there is a saturation problem. An individual cell trajectory can't be too transparent or it can't be seen. This makes the opacity saturate, contracting the upper range of this scale. The second limitation is size. The figure size makes this information difficult to see without enlarging the image on a PDF. The third is that scales that run from white to dark for a single color have a limited amount of information they can convey.

We experimented with other ways to convey this information along with the vector field. They all ran into similar presentation problems. One option is to expand the scale to use several colors. For the figure in the main text we feel that the usefulness of having a consistent color scheme where each history condition is associated with a single color throughout the manuscript outweighs the drawbacks above. However, we have added Expanded View Figure 3 that falls on the other side of this tradeoff and provides more information on the density of the vector field in exactly the way suggested by the reviewer. We have split Figure 2 into two figures (now Figures 3 and 4) and resized it so that this information can be seen a little better.

#3. Reviewer 1. Depth of the "sticky point" and mathematical model: For me, the major weakness of the manuscript was that there was no mathematical model despite the richness of their data in a well characterized system that would make modelling a possibility. The main claim of the authors is that the vector field (Fig. 2) reveals that cells that underwent a Long-Term Glucose Repression (LTGR) are stuck in a "sticky point" that's near the $Gal3p = 0$ and $Gal1p = 0$. But can this qualitative statement be made stronger with a quantitative model? How "sticky" is this place? Perhaps the authors can write down the differential equations for the galactose network (as in Acar et al, Nature 2005) and then approximate the stochastic gene expression using Fokker-Planck approximation. From this Fokker-Planck approximation, one can calculate how long it would take for a cell to escape the "potential well" that might characterize the "sticky location" ($Gal3p \sim 0$, $Gal1p \sim 0$). Admittedly, this is easier said than done and perhaps it's beyond the scope of the current study. But I think some estimation of the depth of the well that traps the cells in the "sticky region" (e.g., by linking the "hours in galactose until expression crosses 10% of plateau level" in Fig. 4B with a model that estimates this depth) would make the claim more quantitative. I don't expect an extensive model but some quantitative reasoning would strengthen the work.

Reviewer 2. A mathematical model capturing the observations (even if simple) would be very useful for the paper. Perhaps there is no need to go into full-detail modeling of the GAL network - a simple model including the activators and their degradation, as well as their effect of cell division or death rates may be sufficient. This would also increase the interdisciplinary relevance of the manuscript.

We considered developing a model for this system both before our initial submission and after reading the reviews. However, while we agree that modeling this system would be useful, we feel it is out of the scope of this paper. Models of molecular networks are primarily useful for describing the system and making experimental predictions about what would happen if we varied parameters or changed the assumptions of the model. Unless we were to do further experiments, making a model and fitting it to our data would tell us little beyond what our data already shows. For example, Figure 2 and Appendix Table S2 already show the actual distribution of escape times from this sticky region, which is the quantitative data that the reviewer requested. To go from these waiting times to some kind of potential (depth of the well) would require not only a model but also more experimental work to validate it. Using pre-existing models, like the Acar *et al.* 2005 model, would require measurements, of Gal80p for example, that we don't have.

We definitely agree that future work on this system would benefit from a model, in particular to serve as an organizing framework to investigate the relationship between energy/resources and induction dynamics. We have shown that this relationship is clearly important, and yet resource constraints are not included in any systems biology type models of this network

that we know about. One problem with incorporating them into a model at this point is that we know so little about the mechanisms by which energy or resources affect the system. The constraints could involve levels stored carbohydrate reserves, cell age, ribosome availability, etc. If we were to develop a model with a 'resource' variable affecting the dynamics, the interactions we would specify would be somewhat arbitrary, making the parameters we would estimate uninformative. We feel that the appropriate approach would be to develop a set of models where energy/resources interact with the network in different ways, and then design specific experiments to distinguish between these. But this would be best done in another study. One important goal of such a research project would be to determine more specifically what this energy/resource component is at the cellular or molecular level. For example, if the bootstrapping is driven by stochastic gene expression, then how, mechanistically, are these wait-times shortened by the extra glucose? Even if we were to develop a model that links these two, we would have no way to rigorously test it with our current data.

#4. *Reviewer 1. Stable / unstable fixed points in the vector field: Indicate where the stable and unstable fixed points are (i.e., fixed point values for Gal3p and Gal10p) in the vector fields (Fig. 2).*

The figure has grid lines indicating expression levels. Instead of marking the fixed points on the figures, we have added a sentence in the text to this effect (page 5, 1st paragraph). One interpretation of our data that we discuss is that each cell has a slightly different absolute stable fixed point (see new Appendix Table S3), and so we felt that specifically placing single points on the vector field would be imply too much.

#5. *Reviewer 1. Some of the labels and vector arrows are too small to see on paper or regular sized PDF: Examples include the vector arrows in Fig. 2 and Fig. S12, the labels for hours and sugar types in Fig. S2, and Fig. S10.*

Reviewer 2. Font size is too small (almost invisible) in Fig. S2 and other figures. Please check font size throughout the manuscript and the SI when this comment is applicable

We have made sure that font sizes follow the MSB figure guidelines.

#6. *Reviewer 1. Some minor typos here and there*

We found some and corrected them.

#7. *Reviewer 2. Throughout the manuscript, GAL1 and GAL3 are called "inducers". However, the standard use of the term "inducer" refers to a small molecule that causes gene expression increase by binding a transcription factor. To avoid this confusion, Gal1p and Gal3p should not be called "inducers". In fact, the sugar galactose is the inducer for the GAL network. Gal1p and Gal3p could perhaps be called "positive regulators".*

The use of the term inducer to refer to Gal3p and Gal1p has a history in the GAL network literature (e.g. Bhat & Hopper 1991), but it is not a standard usage for the larger field, and we agree that this non-standard use of the term is confusing. In addition, we talk about galactose induction and induction dynamics, and these are referring to what happens to the cells after they are placed in galactose, so our usage was inconsistent. The term positive regulator is better, but clunky, and we also do not want to give the impression that these proteins are actively interacting with promoters like a transcription factor. We have decided to use the term *transducer* (after Castillo-Hair *et al.* 2015 and others) to refer specifically to these proteins. A transducer converts a signal from one form to another. In the galactose network, galactose interacts with Gal3p and enables it to associate with Gal80p, thereby freeing Gal4p to do its job. In this way, Gal3p transduces the galactose signal into biological action during GAL network induction. This change has been made in multiple places in the text, and we introduce it in the first paragraph of the introduction.

#8. *Reviewer 2. The concept of the vector field seems to be useful. It would be helpful to illustrate with a cartoon how the field arrows are generated. It would also be useful to be very explicit in describing what the vector field reveals that other visualization approaches could not. Likewise for the variance of the vector field.*

We have prepared a new figure for the Appendix (Appendix Figure S1) that illustrates how the vector field arrows are estimated and how the circular variance is estimated. We have also added text (page 3, paragraph 1) that describes in more detail what a vector field representation is and what kinds of information it is good at conveying. We have also added a new figure for the Appendix that illustrates how cells move on a vector field (Appendix Figure S2).

#9. Reviewer 2. One critical piece of information missing from the vector field plots is time. Would there be a way to indicate time by some faint contour lines that the vectors are perpendicular to? I think this would be useful.

A vector field representation is dynamic in that it represents flows on a state space (in this case how the concentrations of Gal1p and Gal3p change) but time is only implicitly a part of it – the lengths of the arrows represent speed. Different cells may reach the same concentrations of Gal1p and Gal3p at different times in the experiment (particularly for glucose history), but the vector field concept suggests that once a cell is at this point in state space, its future state only depends on its current state and not on how long it took to get there. Our new description of a vector field makes these points (page 3, paragraph 1).

#10. Reviewer 2. What does really happen after LTGR? I would guess that the protein Gal2p must play a key role because it takes up galactose. Can the cells ever induce if GAL2 is deleted? The role of Gal2p should be clarified in a couple of sentences. Maybe the depletion of Gal2p is equally important as the depletion of Gal1p and Gal3p.

Gal2p is the key importer of galactose into the cell, but it too is absent initially after glucose repression and needs to be produced. With Gal2p absent, the role of importing galactose falls to a constitutive facilitated diffusion process that doesn't require GAL induction. *gal2* deletion strains can indeed induce (Hawkins & Smolke 2006) and show an interesting difference from wild-type in the relationship between steady-state induction levels and galactose concentration, but to our knowledge experiments have not been done to investigate its influence on the transient dynamics. Because other transporters can move galactose into the cell and because Gal2p is initially absent, we do not expect it to be crucial for the initial transient dynamic patterns we discuss in this paper. That said, it is Gal2p is a more efficient transporter of galactose and so one might imagine that once it is produced it increases the accumulation of internal galactose more quickly than another transporter would, potentially accelerating induction. We agree that Gal2p and Gal80p as well are important players in this system. However, we decided to focus on Gal1p and Gal3p because previous work had indicated that the presence or absence of these two genes predicted the type of response – slow/fast, bimodal/graded. This data suggested that it could be fruitful to analyze memory-dependent effects and transient induction in terms of these two proteins.

We have added a short discussion of Gal2 in the text (1st paragraph of Results) and also clarified our motivation for choosing to focus on Gal1p and Gal3p. We have also added an additional table in the appendix that summarizes some of these kinds of observations in the literature that led us to do this work. (Appendix Table S1).

#11. Reviewer 2. The term "bootstrapping" is heavily used, but not clearly explained. It is a contracted idiom that may not be familiar to many non-native speakers. Therefore, the Authors should clarify what is the general meaning of bootstrapping in everyday English, and how does the cells' behavior relate to this meaning of the word "bootstrapping"?

We have added a sentence in the text to introduce and define the term when we first use it. (page 3, paragraph 2)

#12. Reviewer 2. The current manuscript lacks a Methods section in the main text. It may be worth considering to move some of the methods to the main text. A Discussion would also be useful.

We have reworked the paper organization so that it conforms to the Research Article format for MSB.

#13. Reviewer 2. The set of image processing Matlab scripts with an example image series should

be made available online as Supplementary Material.

The full dataset along with image and statistical analysis scripts will be available in the UCSD digital collections research data archive (<http://library.ucsd.edu/dc>). A DOI number is forthcoming.

#14. Reviewer 2. After mentioning that "Gal1p is one of the most highly induced genes in yeast - it is upregulated 1000-fold - which makes the GAL1 promoter a useful tool for genetic engineering" it may be useful to cite a couple of references, for example, PMID:27111147 or PMID:19279212. Other useful references may be PMID:24453942 and PMID:25626086.

We have added pertinent references and thank the reviewer for highlighting these examples.

2nd Editorial Decision

19 December 2016

Thank you again for submitting your work to Molecular Systems Biology. We have now heard back from reviewer #1 who agreed to evaluate your study. As you will see below, s/he is satisfied with the modifications made and thinks that the study is now suitable for publication.

Before we formally accept the manuscript for publication, we would ask you to provide the DOI for the dataset deposited in the UCSD Digital Collections.

REFeree REPORTS

Reviewer #1:

I am satisfied with the revised manuscript. I recommend publication.

- The splitting of Figure 2 into Figs 3 and 4 and the addition of expanded view figure 3 to more clearly illustrate the density of vector field helped me better understand the data.

- Absence of mathematical model: I agree that if the authors were to develop a mathematical model, then further experiments would be necessary to validate the data because the model would have to make some predictions of experiments that have not yet been carried out. This would quickly grow the manuscript out of its scope. In lieu of the model, the authors have revised many parts of their manuscript, including the discussion sections, that helped me form a clearer mental picture (qualitative model) of what is causing the growth defects associated with the long-term glucose repression. This is sufficient for me.

- Addition of Appendix Table S3 and revised 1st paragraph on Pg 5: These address my suggestion of indicating the location and stability of fixed points.

- I agree with the authors' response to Reviewer 2's comments and their other revisions.

Hyun Youk.

Corresponding Author Name: Scott A. Rifkin
 Journal Submitted to: Molecular Systems Biology
 Manuscript Number: MSB-16-7323